# A Comparative Analysis of Orthotopic and Subcutaneous Pancreatic Tumour Models: Tumour Microenvironment and Drug Delivery

**DOI:** 10.3390/cancers15225415

**Published:** 2023-11-14

**Authors:** Jessica Lage Fernandez, Sara Årbogen, Mohammad Javad Sadeghinia, Margrete Haram, Sofie Snipstad, Sverre Helge Torp, Caroline Einen, Melina Mühlenpfordt, Matilde Maardalen, Krister Vikedal, Catharina de Lange Davies

**Affiliations:** 1Department of Physics, Norwegian University of Science and Technology, 7491 Trondheim, Norway; sara.b.s.arbogen@ntnu.no (S.Å.); sofie.snipstad@ntnu.no (S.S.); melina@exact-tx.com (M.M.); krister.vikedal@rr-research.no (K.V.); catharina.davies@ntnu.no (C.d.L.D.); 2Department of Structural Engineering, Norwegian University of Science and Technology, 7491 Trondheim, Norway; mj.sadeghinia@ntnu.no; 3Department of Clinical and Molecular Medicine, Norwegian University of Science and Technology, 7491 Trondheim, Norway; karin.margrete.haram@stolav.no (M.H.); sverre.torp@ntnu.no (S.H.T.); 4Cancer Clinic, St. Olavs Hospital, 7006 Trondheim, Norway; 5Department of Radiology and Nuclear Medicine, St. Olavs Hospital, Trondheim University Hospital, 7006 Trondheim, Norway; 6Department of Pathology, St. Olavs Hospital, Trondheim University Hospital, 7006 Trondheim, Norway; 7EXACT Therapeutics, 0581 Oslo, Norway; 8Department of Engineering Science, University of Oxford, Oxford OX1 3NP, UK; 9Department of Microbiology, Oslo University Hospital, 0424 Oslo, Norway

**Keywords:** drug delivery, tumour microenvironment, PDAC animal model, orthotopic, subcutaneous, preclinical model, collagen, collagen structure, vasculature, immune cell infiltration, macrophages, neutrophils, functional vessels

## Abstract

**Simple Summary:**

Pancreatic tumours present significant treatment challenges due to their resistance to chemotherapy and complex tumour microenvironment. Choosing the appropriate preclinical models and understanding how their characteristics affect drug delivery to the tumours is essential for designing clinically relevant experiments. This study investigates pancreatic tumours growing orthotopically or subcutaneously and presents the properties of their tumour microenvironments and their impacts on drug delivery.

**Abstract:**

Pancreatic ductal adenocarcinoma (PDAC) remains a challenging malignancy, mainly due to its resistance to chemotherapy and its complex tumour microenvironment characterised by stromal desmoplasia. There is a need for new strategies to improve the delivery of drugs and therapeutic response. Relevant preclinical tumour models are needed to test potential treatments. This paper compared orthotopic and subcutaneous PDAC tumour models and their suitability for drug delivery studies. A novel aspect was the broad range of tumour properties that were studied, including tumour growth, histopathology, functional vasculature, perfusion, immune cell infiltration, biomechanical characteristics, and especially the extensive analysis of the structure and the orientation of the collagen fibres in the two tumour models. The study unveiled new insights into how these factors impact the uptake of a fluorescent model drug, the macromolecule called 800CW. While the orthotopic model offered a more clinically relevant microenvironment, the subcutaneous model offered advantages for drug delivery studies, primarily due to its reproducibility, and it was characterised by a more efficient drug uptake facilitated by its collagen organisation and well-perfused vasculature. The tumour uptake seemed to be influenced mainly by the structural organisation and the alignment of the collagen fibres and perfusion. Recognising the diverse characteristics of these models and their multifaceted impacts on drug delivery is crucial for designing clinically relevant experiments and improving our understanding of pancreatic cancer biology.

## 1. Introduction

Pancreatic ductal adenocarcinoma (PDAC) presents a poor prognosis with a high mortality rate and a five-year survival rate of less than 10% [1]. The high mortality rate is associated with a late diagnosis at advanced stages and a resistance to chemotherapy [2].

Stromal desmoplasia is a typical histological hallmark of PDAC tumours [3], which describes a dense fibrotic tissue. It is characterised by the proliferation of pancreatic stellate cells and an increased production of extracellular matrix components, such as collagen, fibronectin, and hyaluronan [4]. Activated pancreatic stellate cells are critical contributors to this excessive fibrotic tissue. The resulting dense stroma creates a physical barrier that can lead to high interstitial pressure and limited drug delivery [5,6,7].

This stromal reaction plays a significant role in the pathobiology of pancreatic cancer and is closely associated with a poor prognosis [8]. The dense fibrotic stroma not only hinders effective drug penetration but also promotes the development of a hypoxic and nutrient-deprived tumour microenvironment, fostering cancer cell survival and proliferation [6,9]. The stroma also facilitates the formation of a protective niche for cancer stem cells, which is thought to drive tumour initiation, progression, and therapy resistance. Moreover, stromal desmoplasia is associated with increased invasiveness, lymphatic and vascular invasion, and distant metastasis, further contributing to the poor prognosis of pancreatic cancer. Additionally, the stromal desmoplastic reaction in pancreatic cancer has been shown to have immunosuppressive effects, creating an immunologically cold tumour microenvironment [10].

The stromal desmoplasia’s main component is collagen. Therefore, both the collagen amount and the orientation of the collagen fibres have significant implications for the delivery and distribution of therapeutic agents in pancreatic tumours. The alignment of the collagen fibres can influence the path through which drugs diffuse in the tumour [11].

Various preclinical tumour models have been developed to study PDAC, in addition to in vitro models such as cell lines and organoids [12]. Cell lines are frequently used for high-throughput bioinformatics studies, genetic manipulation, and co-culture experiments to study tumour stroma interactions. Organoids provide a platform for high-throughput drug screening and have been linked to primary tumours in terms of genetic profiles [13,14]. Regarding the tumour models, patient-derived xenografts (PDXs) and genetically engineered mouse models (GEMMs) are commonly used. PDXs are established by transplanting human tumour specimens into immunodeficient mice. They can recapitulate the characteristics of cancer and retain the genomic features of the patients and the intratumour heterogeneity of cancer. GEMMs are widely used because of their simplicity, although the complexity and heterogeneity of the tumour microenvironment and the stromal desmoplasia of PDAC are often underrepresented in GEMMs [12].

Orthotopic and subcutaneous PDAC models are commonly used. The murine KPC model (KrasLSL-G12D/+; Trp53LSL-R172H/+; Pdx-1-Cre) is used to mimic the genetic alterations present in human PDAC [15,16].

Subcutaneous KPC tumours are formed by subcutaneously injecting the tumour cell suspension. This method is frequently used because of its simplicity [17]. The orthotopic model is established by injecting the tumour cells into the pancreas via laparotomy [18]. This model is commonly considered to be more clinically relevant because the tumours grow in the pancreatic environment.

In terms of simplicity, it is generally accepted that the subcutaneous model is superior to the orthotopic pancreatic tumour model [19]. The subcutaneous model is simple to implant. The tumours are easily accessible for measurements, enabling efficient and straightforward tracking for tumour progression and the response to treatment. In contrast, the orthotopic model requires a technically challenging procedure for implantation and is not easily accessible for measurements, as the tumour growth needs to be monitored by magnetic resonance imaging (MRI) or ultrasound imaging. However, the orthotopic tumour is advantageous in mimicking organ-specific conditions, tumour–host interactions, invasion, metastasis, and treatment responses [18,20]. Some findings suggested that the phenotypic properties of metastatic cells were regulated by the expression of the genes activated through interactions with the organ’s environment [21]. This supports the advantages of the orthotopic model for studies that aim to elucidate the molecular events associated with the progression and metastasis of cancer [22], enabling a more comprehensive understanding of tumour biology. In the specific case of PDAC, the results have shown that the orthotopic and heterotopic murine models of PDAC respond differently to treatment with FOLFIRINOX (calcium folinate, fluorouracil, irinotecan, oxaliplatin) [23], the combined chemotherapy that is frequently used for the treatment of pancreatic cancer.

This study compared the subcutaneous and orthotopic KPC tumour models in terms of the tumour growth, histopathology, infiltration of immune cells, functional vasculature, perfusion, and collagen content and organisation. A novel aspect was the broad range of tumour properties that were studied, and especially the extensive analysis of the structure and orientation of the collagen fibres in the two tumour models. The impact of the tumour characteristics on the tumour uptake of a fluorescent macromolecular model drug was assessed to evaluate the suitability of the two tumour models for drug delivery studies.

## 2. Materials and Methods

### 2.1. Mice

The female albino B6 mice were acquired at an age between 6 to 8 weeks (Janvier Laboratorie, Le Genest-Saint-Isle, France). The mice were housed in ventilated cages (IVCs) (Model 1284 L, Techniplast, France) in groups of five and had free access to food, sterile water, and a controlled environment with temperatures between 19 and 22 °C and a relative humidity between 50% and 60%. The mice were treated following the recommendations from the Federation of European Laboratory Animal Science Associations (FELASA), and all the experimental animal procedures complied with the protocols approved by the Norwegian National Animal Research Authorities.

### 2.2. Tumour Cell Line and Tumour Implantations

The murine pancreatic cell line KPC001S gLuc/green fluorescence protein was generously provided by Steele Laboratories, Massachusetts General Hospital, Harvard University. The cells were cultured in Dulbecco’s modified Eagle medium (Gibco Invitrogen, Carlsbad, CA, USA) with 10% fetal bovine serum and 1% penicillin–streptomycin (both from Sigma Aldrich, St. Louis, MO, USA) at 37 °C and 5% CO_2_.

For the subcutaneous implantation of the tumour cells, the mice were anaesthetised using inhaled 2% isoflurane in 40% O_2_ and 60% NO_2_ (Baxter AS, Oslo, Norway). A 20 μL suspension containing 200,000 KPC cells was slowly injected subcutaneously into the lateral aspect of the left hind leg between the hip and the knee. During the implantation, the body temperatures of the mice were maintained at a constant level using a heating pad.

For the orthotopic implantation of the tumour cells, the mice were anaesthetised using inhaled isoflurane and local anaesthesia in the incision area with 0.04 mL of 10 mg/mL lidocaine (Accord Healthcare Limited, Middlesex, UK). Analgesia was provided through a subcutaneous injection of 0.07 mL of buprenorphine 0.3 mg/mL (Indivior Europe Limited, Dublin, Ireland) and 0.1 mL of meloxicam 5 mg/mL (Boehringer Ingelheim, Rohrdorf, Germany). The surgical area was shaved using an electric clipper and shaving cream (Veet, Reckitt Benckiser Healthcare, Hull, UK), and the surgical area was sterilised with chlorhexidine 5 mg/mL (Fresenius Kabi, Halden, Norway). The cells were implanted via laparotomy [24,25]. A subcostal incision of 5–7 mm was made on the skin to access the left side of the abdominal cavity. The peritoneum was incised with a 5 mm incision. The spleen and pancreas were externalised using precision forceps, and a 20 μL suspension containing 200,000 KPC cells in a growth medium was injected into the tail of the pancreas towards the body. The injection angle was kept parallel to the longitudinal axis of the pancreas. The spleen and pancreas were put back into the peritoneal cavity, and the peritoneum was sutured with three surgical knots using Vicryl 6.0 resorbable sutures (Ethicon, Somerville, NJ, USA). The skin was closed using EZ metal clips (Stoelting Co., Wood Doyle, IL, USA). During the implantation, the body temperature of the mice was maintained at a constant level using a heating pad and a heating lamp. After the procedure, the mice were kept in a recovery chamber overnight with a constant temperature of 28 °C.

Both tumour models were allowed to grow for 14–15 days until they reached a diameter of approximately 8–10 mm.

### 2.3. Volume Growth Curves and Metastases

The volumes of the orthotopic tumours were obtained from 3D ultrasound scans using the Vevo3100 imaging system and the broadband cardiac probe MX550 D with a 40-MHz centre frequency (FUJIFILM, Visualsonics, Toronto, ON, Canada) and a 3D step size of 0.076 mm. To measure the tumour volumes, the 3D scans were imported into VevoLab (FUJIFILM Visualsonics). The tumours were visualised and their boundaries were segmented by drawing regions of interest (ROI) every 0.15 mm. The software generated a 3D volume representing the tumour and estimated the volume based on the segmented data. The dimensions of the subcutaneous tumours, including the two perpendicular diameters and the height, were measured using a calliper to calculate the growth curves. Additionally, B-mode 3D scans were obtained for the subcutaneous tumours on the last day of the experiment. The tumours, both orthotopic and subcutaneous, were excised, weighed, and measured. The mice were systematically evaluated for infiltration and metastatic spread of the tumors to adjacent tissues and organs through surgical examination.

### 2.4. Contrast-Enhanced Ultrasound Imaging and Analysis

Contrast-enhanced ultrasound (CEUS) imaging was used to visualise the blood flow in the tumours. A total of 50 µL of the contrast agent MicroMarker (FUJIFILM, Visualsonics, ON, Canada) was injected intravenously, which enhanced the ultrasound contrast and enabled the visualisation of the blood flow. The imaging was performed using the Vevo3100 system and the cardiac probe MX250 (FUJIFILM, Visualsonics, ON, Canada). The imaging was performed in a nonlinear contrast mode, and the frequency used was 18 MHz. The Vevo CQ module in VevoLab was used to estimate the perfusion from the CEUS analysis. The software analysed the contrast agent’s dynamics and calculated the various perfusion parameters based on the user-selected ROIs of the tumours. The following parameters were obtained from the analysis. PE referred to the peak enhancement, representing the maximum contrast agent intensity. WiAUC denoted the wash-in area under the curve, quantifying the contrast agent accumulation over time until the PE. RT stood for the rise time, indicating the time needed to reach the PE. To visualise the distribution of MicroMarker in the tumour, VevoLab was used to create a maximum intensity projection (MIP) from the nonlinear contrast mode recordings taken during the bolus injection.

### 2.5. Tumour Sectioning for Microscopy

The mice were euthanised by cervical dislocation on days 14 or 15 after the tumour implantation. The criterion was similarly sized tumours within and between the subcutaneous and orthotopic groups. The tumours were embedded in Tissue-Tek^®^ O.C.T. (Optimal cutting temperature) (Sakura, Alphen aan den Rijn, The Netherlands) and mounted onto a cork base before freezing in liquid nitrogen. Frozen tumour sections of 25 μm were cut.

### 2.6. Histopathology

The frozen tumour sections were stained with hematoxylin–erythrosine–saffron (HES); hematoxylin, erythrosine (both from Sigma Aldrich), Safran (VWR Corp., Radnor, PA, USA), and Masson trichrome (MT) to differentiate between the collagen and muscle tissues, showing the collagen in blue and the muscles in red. A senior pathologist evaluated the sections. The sections were imaged using bright field microscopy on the Zeiss LSM 800 (Zeiss, Oberkochen, Germany). A 20×/0.8 Plan Apochromat air objective captured tile scans of the entire section.

### 2.7. Confocal Laser Scanning Microscopy of the Functional Vessels

To label the functional tumour vasculature, 50 µL of fluorescein-isothiocyanate (FITC)-labelled Lycopersicon esculentum tomato lectin (2 mg/mL; Vector Laboratories, Peterborough, UK) was injected via a tail vein catheter 5 min before euthanasia. To image the functional vasculature, the frozen tumour sections (25 μm thick) were mounted with Vectashield (Vector Laboratories, Peterborough, UK) and covered with a cover glass before imaging using the Leica TCS SP8 MP (Leica TCS SP8, Leica Microsystems, Mannheim, Germany) by confocal laser scanning microscopy (CLSM) with a HC PL APO 20× dry lens and a numerical aperture of 0.75. For each tumour, images were acquired from one to two tumour sections, with a total of eight to 25 images obtained for each section. The images had a frame size of 1024 × 1024 pixels, an 8-bit format, and a pixel size of 443.8 nm. Excitation was carried out using a 488 nm diode laser, and the emission detection was between 495 and 540 nm.

The CLSM images were analysed using ImageJ (version 1.51j). The images from the FITC-detecting channel were thresholded using the Otsu algorithm, and the percentage of pixels was estimated from the binary image. This gave quantitatively the pixel counts corresponding to the functional vessels. To derive a representative metric for each tumour, we averaged the number of FITC-positive pixels across all the images taken for that tumour.

### 2.8. Collagen Imaging using Second-Harmonic Imaging Microscopy

The distribution of the collagen fibres in the frozen tumour tissue was imaged using second-harmonic imaging microscopy (SHIM) [26]. A titanium–sapphire (Ti: Sp) Chameleon Vision-S two-photon laser (Coherent Inc., Santa Clara, CA, USA) at 890 nm mounted on the Leica SP8 CLSM was used. The second-harmonic generated (SHG) signal was detected both in the forward and backward directions. A 445 ± 10 nm emission filter was placed in front of the two collagen detectors. An air objective of 20× with a numerical aperture of 0.75 was used. For each tumour, images were acquired from one to two tumour sections, with a total of eight to 25 images obtained for each section.

The images were analysed using the ImageJ software. The channel detecting the forward SHG signal was chosen to assess the amount of collagen. An Otsu algorithm was applied to threshold the image, generating a binary image. The percentage of pixels with a SHG signal was measured to quantify the amount of collagen fibres.

The ratio of the forward signal divided by the backward signal (F/B) ratio, pixel by pixel, gave information about the structural organisation and alignment of the collagen fibres. All the pixels in the resulting image that had a value under one were excluded, and the mean intensity of the remaining pixels was measured. The mean intensity of all the images from each tumour section was averaged to obtain a representative value for each tumour. A high F/B ratio indicated well-oriented and aligned collagen fibres, with a stronger forward-propagating SHG signal than the backward-propagating signal.

### 2.9. Collagen Anisotropy Analysis

A customised MATLAB (The MathWorks, Natick, MA, USA) script was developed to assess the dispersion of the collagen fibres in the SHG images, following a previously described method [27]. It employed Fourier transformation and wedge filters to generate a probability density function reflecting the distribution of the collagen fibres. Initially, a 2D Tukey window was applied to the image. Subsequently, the image underwent Fourier transformation and was multiplied by its conjugate complex, yielding the power spectrum density. This step enabled the identification of the fibre direction based on the frequency and orientation. To extract the fibre orientations at specific angles θ, a wedge-shaped filter spanning −89° to 90° with a 1° increment was employed. The relative amplitude of the fibre distribution was determined by this filter. To smooth the data, a moving average filter with a 7° range was applied to the angle θ [27]. The distribution was fitted by two families of fibres using a von Mises distribution.
(1)$$\rho_{\mathrm{vm}}\left(\theta \right)=w\frac{1}{\pi }\frac{\exp\left\{a_{1}\cos\left[2\left(\theta -\alpha_{1}\right)\right]\right\}}{I_{0}(a_{1})\;}+(1-w)\frac{1}{\pi }\frac{\exp\left\{a_{2}\cos\left[2\left(\theta -\alpha_{2}\right)\right]\right\}}{I_{0}(a_{2})\;}$$
where $\rho_{\mathrm{vm}}\left(\theta \right)$ is the von Mises distribution, characterised by the mean fibre angle (α) and concentration parameter (a). The subscript denotes the first or second fibre family. w is the weighting factor with values from 0 to 1 and $I_{0}$ is the zero-order modified Bessel function of the first kind. An average fibre concentration parameter was defined to represent the degree of anisotropy for each image.
(2)$$\bar{a}=a_{1}w+a_{2}(1-w)$$
where the higher the $\bar{a}$, the higher the concentration parameter, indicating more aligned collagen fibres.

### 2.10. Stiffness Measurements by Macro-Indentation

The tumour stiffness was measured using macro-indentation with a custom indenter. The whole frozen tumours were thawed and compressed at 200–400 μm with a speed of 2 μm/s using a spherical indenter with a 0.5 mm radius at four to five locations on the tumour surface. Young’s modulus was estimated by fitting the Hertz model following Equation (3) to the first 200 μm of the experimental force-indentation curves [28].
(3)$$F=\frac{4}{3}\frac{E}{\left.{\left(1-\nu \right.}^{2}\right)}R^{\frac{1}{3}}\delta^{\frac{3}{2}}$$
where E is Young’s modulus, F is the indentation force, R is the radius of the indenter, δ is the indentation depth, and ν is the Poisson ratio, which was assumed to be 0.5. The individual Young’s modulus estimates were averaged to find the overall stiffness per tumour. Five orthotopic and ten subcutaneous tumours were used for indentation.

### 2.11. Immunostaining and Flow Cytometry

Single-cell suspensions of the subcutaneous and orthotopic tumours were prepared by enzymatic disintegration. The tumours were cut into pieces and placed in a solution of 86.2 μL of liberase DL (13 U/mL) (Roche, Vienna, Austria), 86.2 μL of liperase TL (13 U/mL) and 55 μL of DNAse (150 U/55 μL) (Qiagen, Diagnostica, Hillerød, Denmark) in a 4 mL PBS and incubated at 37 °C for 65 min under continuous rotation. The disintegration was stopped by adding a 10 mL PBS with 1% bovine serum albumin (BSA) (Sigma-Aldrich). The cell suspension was filtered and centrifuged at 1500 rpm for 5 min. The immune cells were stained by direct immunostaining after a blocking step using mouse seroblock FcR (BioRad, Herculus, CA, USA) for 15 min at 4 °C. The antibodies used included the following. Anti-CD11b-Alexa Fluor 488 (Biolegend, San Diego, CA, USA) stained mostly innate cells, anti-F4/80-Alexa Fluor 647 (Biolegend) stained macrophages, and anti-Ly6G-Brilliant Violet 421 (Biolegend) stained neutrophils. The immune cells were incubated in a mixture of the three antibodies for 60 min. The mixture was prepared by adding 0.5 μL of 0.5 mg/mL, 0.5 μL of 0.2 mg/mL, and 1.25 μL og 0.5 mg/mL antibodies, respectively, from the stock solution to a 100 μL PBS with 1% BSA. The unbound antibodies were removed by washing in the PBS. The dead cells were stained using the Fixable Red Dead Cell Stain Kit (Thermo Fisher Scientific, Waltham, MA, USA), adding 1 μL to 1 mL of 10^6^ cells and incubating for 30 min at 4 °C.

The cells were analysed on a flow cytometer (Gallios Beckman Coulter, Brea, CA, USA) using laser lines of 405 nm, 488 nm, 561 nm, and 633 nm to excite the Brilliant Violet, Alexa Fluor 488, dead cells, and Alexa Fluor 647, respectively. The cells stained with a single fluorochrome were used to compensate for the fluorescence bleed through into the various detectors. Doublettes and aggregates of the cells were removed by gating the forward light scatter signal height versus the forward light scatter signal area. Live single cells were obtained by gating the live/dead fixable red stained cells versus the side scatter signal area. From the gated live cells, the CD11b-positive cells were determined by gating CD11b versus the side scatter signal area. The fraction of macrophages and neutrophils were determined by gating, respectively, CD11b versus F4/80 and CD11b versus Ly6G.

### 2.12. Near-Infrared Whole Animal Fluorescence Imaging

To compare the drug uptake in the orthotopic and subcutaneous tumours, the pegylated macromolecule labelled with an infrared dye (800CW™ PEG, LiCor Biosciences Ltd., Lincoln, NE, USA) was used as a model drug. A tail vein injection of 50 μL, 1 nmol of 800CW™ PEG was followed by imaging (Pearl Impulse Imager, LI-COR Biosciences Ltd., USA). The 800CW™ PEG had a molecular weight in the range of 25–60 kDa, which was within the same order of magnitude as the molecular weight of the protein-based chemotherapeutic agents. The excitation/emission was 785/820 nm. A white light image and a near-infrared (NIR) image were recorded immediately after the injection of the NIR dye, as well as 1 h, 2 h, 6 h, and 24 h after injection. The images were analysed using ImageJ. ROIs were selected around the tumour, quantified for the mean fluorescence intensity, and the resulting values at each time point were normalised to the mean intensity immediately after the injection of the NIR macromolecule. For the subcutaneous tumours, ROIs were drawn to exclude the lateral saphenous vein in the leg of the mice. In the case of the orthotopic tumours, the ROIs were intended to exclude the signals from adjacent organs. The white light image was used to localise the pancreas by identifying the scar from the implantation.

### 2.13. Statistics

All the datasets were analysed using GraphPad Prism v8.0 (GraphPad Software, San Diego, CA, USA). The statistical test depended on the type and number of datasets that were compared and was specified in the caption of the corresponding figure. A *p*-value smaller than 0.05 was considered to indicate significance.

## 3. Results

### 3.1. Histopathological Characterisation

The orthotopic tumours (Figure 1a) showed a poorly differentiated adenocarcinoma with the tumour cells growing in sheets and cords with focal abortive glandular and ductal structures (Figure 1c). The tumour cells infiltrated or replaced normal pancreatic tissue and showed massive infiltration in the adjacent soft tissues, including skeletal muscle. The tumour cells showed marked cytological atypia and a high mitotic activity, and the tumours contained many apoptotic cells. There were several small focal areas with tumour necroses. The desmoplastic response was poor. A moderate lymphocytic infiltration was seen in the periphery, whereas this was sparse in the central tumour areas.

The subcutaneous tumours had large infarct-like necrotic areas, mainly in the centre of the tumours (Figure 1b,d). A more undifferentiated carcinoma was seen and there was infiltration in the skeletal muscle. The lymphocytic response was comparable to that in the orthotopic tumours. The MT-stained sections showed a qualitative increase in the collagen in the subcutaneous tumours compared to the orthotopic ones (Figure 1b,d). Additional images are presented in Appendix A.

### 3.2. Tumour Growth and Metastases

The B-mode ultrasound images of the orthotopic and subcutaneous tumours are presented in Figure 2a,b. Additional images are presented in Appendix A. The orthotopic tumours were primarily localised within the pancreas. However, a prominent protrusion often infiltrated and extended outside the abdominal wall (Figure 2a). This could have resulted from the needle leaking tumour cells as it was retracted from the pancreas during implantation. Additionally, potential damage to the peritoneal wall and surrounding tissue during the surgical procedure may have facilitated tumour growth, forming the protrusion.

In terms of the infiltration and metastatic sites, the orthotopic tumours presented infiltration to neighbouring tissues and were metastasised into several organs: the spleen, intestines, kidneys, and liver, as summarised in Table 1. The subcutaneous tumours presented infiltration into the adjacent muscle tissue, and no metastases were found. Comparing the tumour growth revealed that the orthotopic tumours reached larger volumes, particularly during the final five days of the study period (Figure 2c). Fourteen days after implantation, the orthotopic tumours had an average volume that was two times larger than the subcutaneous tumours (Figure 2d). The final volume of the subcutaneous tumours presented less variability than the orthotopic tumours.

### 3.3. Collagen Fibre Estimation using Second-Harmonic Imaging Microscopy

The SHIM images of the orthotopic and subcutaneous tumours are presented in Figure 3a,b. Additional images are presented in Appendix A. The images from the orthotopic tumours (Figure 3a) showed the presence of potential pancreatic ducts surrounded by collagen lining. The diameter of the lumen of the duct was between 100 and 200 µm. This corroborated the ductal structures observed in the MT-stained sections (Figure 1c). The SHIM analysis showed that the subcutaneous tumours had a significantly higher amount of collagen compared to the orthotopic tumours (Figure 3c), and the collagen fibres seemed to be straighter and more aligned. A quantitative analysis of the F/B intensity ratio showed that the subcutaneous tumours had a significantly higher ratio compared to the orthotopic tumours (Figure 3d).

### 3.4. Structural Organisation and Alignment of the Collagen Fibres

The F/B ratio of the collagen fibres showed a six-time increase in the subcutaneous tumours compared to the orthotopic tumours (Figure 3d). This was in line with the collagen anisotropy analysis (Figure 4), where the subcutaneous tumours presented a trend towards a higher collagen anisotropy concentration parameter, indicating that the collagen fibres were less ordered in the orthotopic KPC tumours.

### 3.5. Tumour Biomechanical Assessment Ex Vivo

To characterise the biomechanical properties, the stiffness was measured in the orthotopic and subcutaneous tumours. The indentation curves for each individual measurement of the orthotopic and subcutaneous tumours are shown in Figure 5a. Young’s modulus was obtained by fitting the Hertz model (Equation (3) to the individual force-indentation curves. The subcutaneous tumours had a significantly higher Young’s modulus compared to the orthotopic tumours (Figure 5b). However, there was a clear intratumour variability (Figure 5a). This suggested that even within the individual tumours, the stiffness was not uniformly distributed.

### 3.6. Functional Vessel Density Analysis from Confocal Laser Scanning Microscopy

The CLSM images of the functional vessels (lectin-FITC-labelled vessels) of the orthotopic and subcutaneous tumours are presented in Figure 6a,b. A quantitative analysis of the number of fluorescent pixels showed a trend towards a higher number of functional vessels in the orthotopic tumours compared to the subcutaneous tumours (Figure 6c). However, this difference was not statistically significant. Notably, there was substantial variability in the number of functional vessels. This was clearly seen in the additional images presented in Appendix A.

### 3.7. Perfusion Analysis from CEUS Imaging

CEUS imaging was used to analyse the perfusion in the two tumour models. The time–intensity curves for the subcutaneous and orthotopic tumours are presented in Figure 7a, showing a higher intensity for the subcutaneous tumours. The PE was more than 2.5 times higher for the subcutaneous tumours compared to orthotopic ones, indicating a greater vascular volume (Figure 7b, Table 2). In accordance with this, the WiAUC was higher for the subcutaneous tumours, showing a larger blood volume in the subcutaneous tumours than in the orthotopic ones. The RT, which indicates the rate at which the contrast agent fills the vascular network, was higher for the subcutaneous tumours, which indicated slower blood flow (Figure 7d, Table 2).

The B-mode ultrasound images and the maximum intensity projection (MIP) of the nonlinear contrast mode images of the orthotopic and subcutaneous tumours are presented in Figure 8a,b, respectively. A qualitative assessment of the images indicated that there were large dark regions with no signal from the contrast agent in the subcutaneous tumours, which were consistent with the larger necrotic area observed in the HES and MT sections (Figure 1d).

### 3.8. Infiltration of Macrophages, Neutrophils, and Innate Immune Cells

The flow cytometry analysis of the orthotopic and subcutaneous KPC tumours showed that both tumour models exhibited comparable levels of infiltrated macrophages, neutrophils, and innate immune cells (Figure 9). The CD11b-positive cells included neutrophils, monocytes/macrophages, dendritic cells, natural killer cells, and subsets of T and B cells. There was a modest trend towards the subcutaneous tumours having a higher presence of neutrophils, macrophages, and innate immune cells. However, there was no statistically significant difference. There was a large difference in the sample size, with the orthotopic group being substantially smaller than the orthotopic group.

### 3.9. Kinetics of Accumulation of 800CW

The accumulation of the macromolecule 800CW in the two tumour models was compared. There was a statistically significantly higher accumulation in the subcutaneous model compared to the orthotopic model (Figure 10a–c). The subcutaneous tumours showed a faster accumulation of 800CW, particularly within the initial 2 h, compared to the orthotopic ones.

The correlation between the uptake of 800CW 24 h after injection, the PE, and the weight of the orthotopic and subcutaneous KPC tumours (Figure 11) was studied. The multi-variable plot showed that for the orthotopic tumours, the smallest tumour presented the highest 800CW uptake and the highest PE (Figure 11a). However, this was based on only one tumour. A negative correlation between the tumour weight and uptake of CW800, and tumour weight and PE were seen, and the Pearson’s coefficients were, respectively, −0.63 and −0.88. The uptake of CW800 and the PE showed a positive correlation with a Pearson coefficient of 0.84. For the subcutaneous tumours, the larger tumours showed a higher uptake of 800CW (Figure 11b). The Pearson’s coefficient for the tumour weight versus the uptake of CW800 was 0.78. A negative correlation was found between the PE and tumour weight, and the PE and uptake of CW800, with Pearson’s coefficients of −0.64 and −0.45, respectively.

### 3.10. Summarised Results

A summary of the results from all the parameters analysed for both tumour models is presented in Table 3.

## 4. Discussion

The histopathological assessment demonstrated the major differences between the orthotopic and subcutaneous tumours. The orthotopic model showed glandular structures that were typical of pancreatic tissue and tumour cells for forming ductal structures and infiltrating healthy pancreatic tissue, which was in line with the prior studies [29]. In contrast, the subcutaneous model did not present ductal structures. The growth rate was higher for the orthotopic tumours, and they reached larger volumes, which was in accordance with the previous studies [23]. The final volume of the subcutaneous tumours presented less variability than the orthotopic tumours, indicating a higher reproducibility of the subcutaneous tumour implantation, as discussed in the previous studies [30]. Additionally, the orthotopic model invaded the adjacent tissues and metastasised into the spleen, kidneys, intestines, and liver. These findings suggested that the orthotopic tumour model mimicked the human PDAC to a larger extent compared to the subcutaneous. Several studies proposed that orthotopic tumour models are preferred because of their tissue site-specific pathology and because they allow for metastasis studies, thus resembling natural tumourigenesis in humans and being more clinically relevant [31]. Additionally, our finding of a network of collagen fibres surrounding the pancreatic ducts in the orthotopic tumours showed the complexity of the orthotopic model and the possible interactions with the stromal cells producing the extracellular matrix.

The microenvironment differed between the orthotopic and subcutaneous tumours, and the type of necrosis was one of the main differences. The orthotopic tumours presented focal necrotic regions, while the subcutaneous tumours had large infarct-like necrotic areas. The MIP images of the subcutaneous tumours also demonstrated larger dark regions with no contrast agent, indicating necrotic areas. This was consistent with the large necrotic regions observed in the histological assessment, as seen in the other subcutaneous PDAC models [32]. The difference in the number of functional vessels and the perfusion might explain the difference in necrosis. Although the CEUS data indicated that the subcutaneous tumours were overall better vascularised than the orthotopic tumours, the lower number of functional vessels imaged by the CLSM might suggest that parts of the tumours had fewer blood vessels, causing larger necrosis. It is essential to add that the CLSM data was based on the microvasculature from a limited number of images of one to two sections from each tumour. Therefore, it was a limited estimation of the fraction of functional vessels in the whole tumour volume. It should also be mentioned that the difference between the orthotopic and subcutaneous tumours regarding the functional vasculature was not statistically significant. The CEUS perfusion analysis was based on a significantly thicker cross-section of the tumour. A notable difference between the CLSM of the functional vessels and the CEUS estimation of the vascular parameters was the spatial resolution, and thus possibly the kind of vessels that were analysed.

The discrepancy in the fraction of functional vessels and the perfusion characteristics between the two models could be explained by the tumour microenvironment and the nature of the blood vessels. The orthotopic tumours had a pre-existing vasculature from the pancreas, which might not have been as effectively branched or distributed as the newly formed tumour vessels in the subcutaneous model, potentially leading to a reduced perfusion [33]. Furthermore, the different anatomical locations had different blood flow rates. It has been argued that this could be due to varying microcirculation across the tissues [34]. Moreover, in the orthotopic model, the tumour cells and associated stromal cells released vasoactive substances that could change the diameter of blood vessels, thus potentially affecting the perfusion [35].

The collagen composition and structure can influence various tumour behaviours, such as growth, invasion, metastasis, and drug resistance and delivery [36,37]. This study found that the subcutaneous tumours presented a higher amount of collagen compared to the orthotopic ones. This could suggest that the subcutaneous tumours showed more stromal desmoplasia, which was in accordance with a previous study [23]. Previous studies have also shown that there might be an inverse relationship between PDAC desmoplasia and tumour vascularity [23,38]. The excessive collagen deposition in the desmoplastic stroma can exert physical pressure on the blood vessels, leading to their compression [39]. This compression could explain the decreased amount of functional microvasculature that was observed in the subcutaneous model.

The biomechanical evaluation of the tumours indicated a significant increase in Young’s modulus in the subcutaneous tumours, suggesting a stiffer tumour microenvironment. The increased stiffness in the subcutaneous tumours corroborated the higher collagen content, which was consistent with other studies [40,41]. Both factors have been used as predictors of tumour aggressiveness [41].

The F/B ratio of the SHG signal reflected the collagen structure and alignment, and the high ratio in the subcutaneous tumours indicated aligned collagen fibres, with a stronger forward-propagating SHG signal than the backward-propagating signal. These findings were in line with the anisotropy analysis, where a moderate trend towards more aligned and ordered collagen fibres in the subcutaneous tumours was observed, whereas the orthotopic tumours showed less collagen anisotropy and a more random distribution of collagen fibres. The more complex interaction between the stromal and tumour cells in the orthotopic tumours might have caused less ordered collagen fibres, as was also seen by the circular organisation of the collagen fibres around the pancreatic ducts.

The flow cytometry analysis of the immune cells infiltrating both tumour models showed comparable levels of neutrophils, macrophages, and innate immune cells. A slight trend towards higher levels of these cell populations was seen in the subcutaneous tumours. A potential increased infiltration of immune cells could have been related to the increased stromal desmoplasia observed in the subcutaneous tumours. Previous studies have found that the fibrotic stroma played an essential role in the production of inflammatory factors and the infiltration of immune cells in PDAC tumours [42].

The difference in the accumulation of the macromolecule 800CW in the two tumour models emphasised the complex interaction between the histopathology, functional vasculature, perfusion, collagen composition and structure, and drug uptake. From our analysis, the subcutaneous model showed a higher and faster uptake than the orthotopic model. The increased accumulation of 800CW in the subcutaneous tumours could be attributed to the enhanced perfusion observed in this model. The higher uptake of 800CW aligned with the increased inflow of the microbubble and a larger overall vascular volume estimated from the perfusion analysis. It could also be that the rapidly formed vasculature in the subcutaneous model was leakier than the pre-existing vasculature in the orthotopic model. Another factor that could explain the increased uptake was the organisation of the collagen fibres. While the subcutaneous tumours showed more collagen than the orthotopic ones, our findings suggested that the ordered and aligned collagen fibres in the former facilitated drug diffusion and uptake [43,44].

A limitation concerning CEUS and whole animal fluorescence imaging was the attenuation due to the absorption and scattering of the ultrasound wave and excitation and emitted light, respectively. The distance from the skin to the orthotopic and subcutaneous tumours was almost the same, as the orthotopic tumours had a protrusion through the abdominal wall that reached towards the skin. Therefore, the difference in the CEUS parameters and tumour uptake of CW800 cannot be explained by attention.

This paper presented important different properties of the tumour microenvironment when PDAC grows orthotopically or subcutaneously, and it highlighted specific characteristics that may be relevant for drug delivery studies. Choosing orthotopic or subcutaneous tumours will depend on the aim of the study. Orthotopic tumours will be the choice for investigating interactions between tumour cells, stromal and host cells, and studying invasive and metastatic behaviours. Subcutaneous tumours have advantages for studying the tumour uptake of macromolecules, showing a higher reproducibility in terms of the tumour volume. Additionally, it demonstrated an enhanced vascular perfusion and a more organised and aligned collagen network, both of which contributed to a resulting increased uptake.

The simplicity of subcutaneous tumours, both in terms of implantation and monitoring tumour growth, makes this model useful for screening the tumour uptake of therapeutic agents and comparing the effects that various treatments might have on drug uptake in tumours and organs [45].

## 5. Conclusions

This study highlighted some significant differences between the orthotopic and subcutaneous tumour models relevant to drug delivery studies. The examination of the tumour growth, histopathology, infiltration of immune cells, functional vessels, perfusion, and particularly the structure and orientation of collagen fibres unveiled a multifaceted insight into their collective impact on the uptake of the fluorescent model drug, 800CW. The orthotopic models more faithfully recapitulated the histopathological features and complexity of the PDAC microenvironment, while the subcutaneous models may offer advantages for drug delivery studies based on their reproducibility, their simplicity, their perfusion characteristics, and the organisation of its collagen fibres. Based on these findings, it appears that the main factors contributing to the difference in the uptake of 800CW between orthotopic and subcutaneous tumours are the orientation of the collagen fibres and the perfusion characteristics. Recognising these models’ diverse characteristics and multifaceted impacts on drug delivery is crucial for designing clinically relevant experiments and improving our understanding of pancreatic cancer biology.

## Figures and Tables

**Figure 1 cancers-15-05415-f001:**
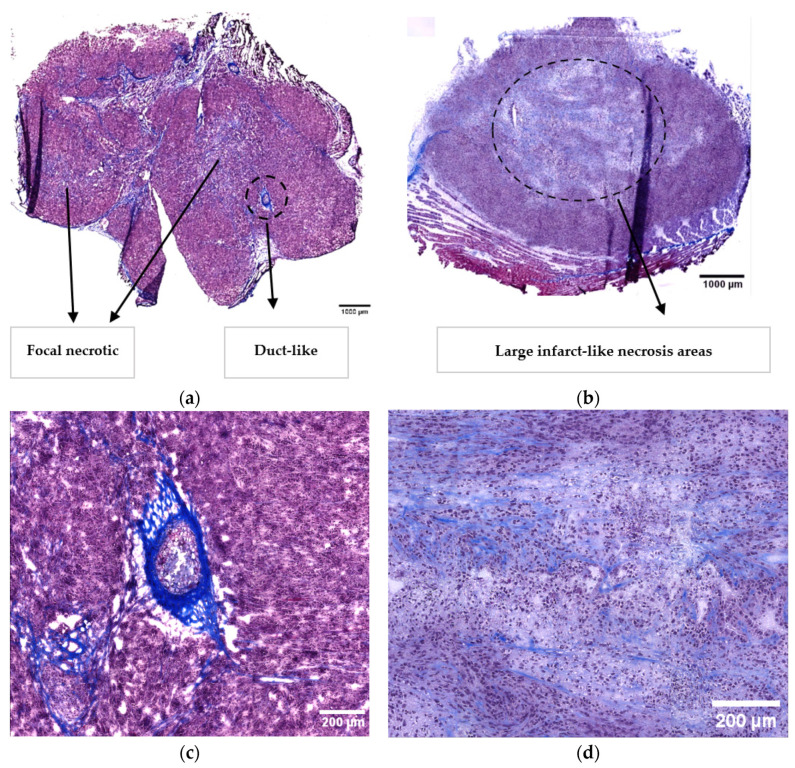
Representative Masson trichrome (MT) images of the orthotopic (**a**) and subcutaneous (**b**) KPC tumours. Collagen is stained in blue. The scale bar represents 1000 μm. Zoomed-in views of the circular areas outlined by dotted lines are shown in panels (**c**,**d**), highlighting a duct-like structure (**c**) and a large infarct-like necrotic region (**d**). The scale bar represents 200 μm.

**Figure 2 cancers-15-05415-f002:**
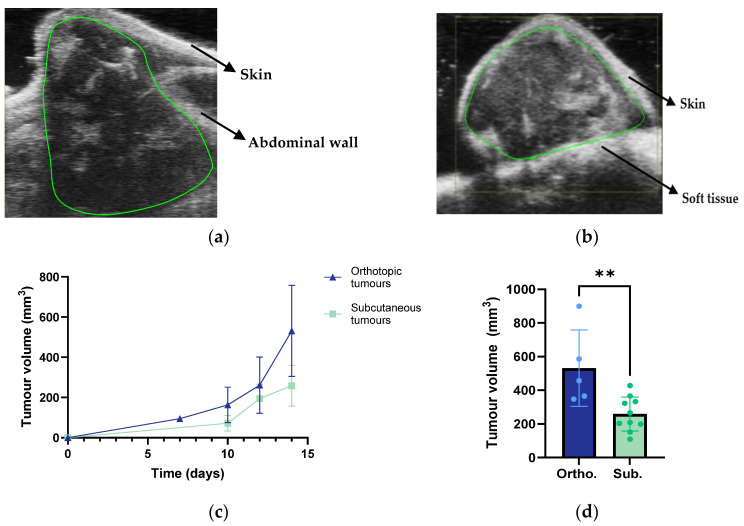
Representative B-mode images and regions of interest (ROI) of the orthotopic (**a**) and subcutaneous (**b**) KPC tumours. Growth curves of the orthotopic and subcutaneous KPC tumours (**c**). Tumour volume 14 days after implantation (**d**). Each dot represents one tumour. The error bars indicate the SD. The asterisks indicate a statistically significant difference evaluated by a two-tailed *t*-test. (** *p* < 0.005). The sample size was *n* = 5 for the orthotopic tumours and *n* = 10 for the subcutaneous group.

**Figure 3 cancers-15-05415-f003:**
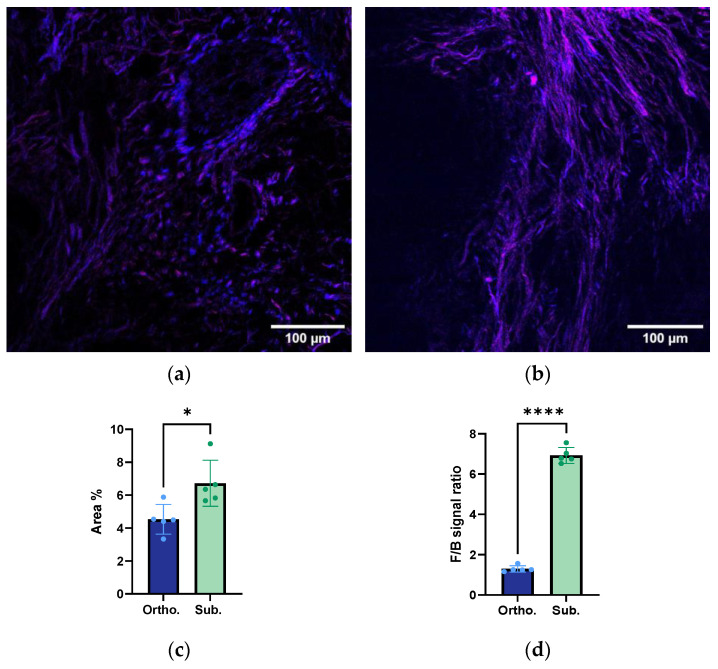
Representative second-harmonic imaging microscopy (SHIM) images of the orthotopic (**a**) and subcutaneous (**b**) KPC tumours. Blue represents the forward second-harmonic generation (SHG) collagen signal and magenta represents the matching backward signal. The scale bar represents 100 μm. The quantitative analysis of the collagen fibres imaged using SHIM of the orthotopic and subcutaneous KPC tumours (**c**,**d**). The total amount of the SHG signal (**c**), with a *p*-value of 0.0186 and the forward/backwards (F/B)-intensity ratio (**d**), with a *p*-value below 0.0001. Each dot represents one tumour. The height of the bar is the mean, and the error bar represents the SD. The asterisks indicate statistically significant differences evaluated by a two-tailed *t*-test. The sample size was *n* = 5 for the orthotopic tumours and *n* = 5 for the subcutaneous group. * *p* < 0.005, **** *p* < 0.0001.

**Figure 4 cancers-15-05415-f004:**
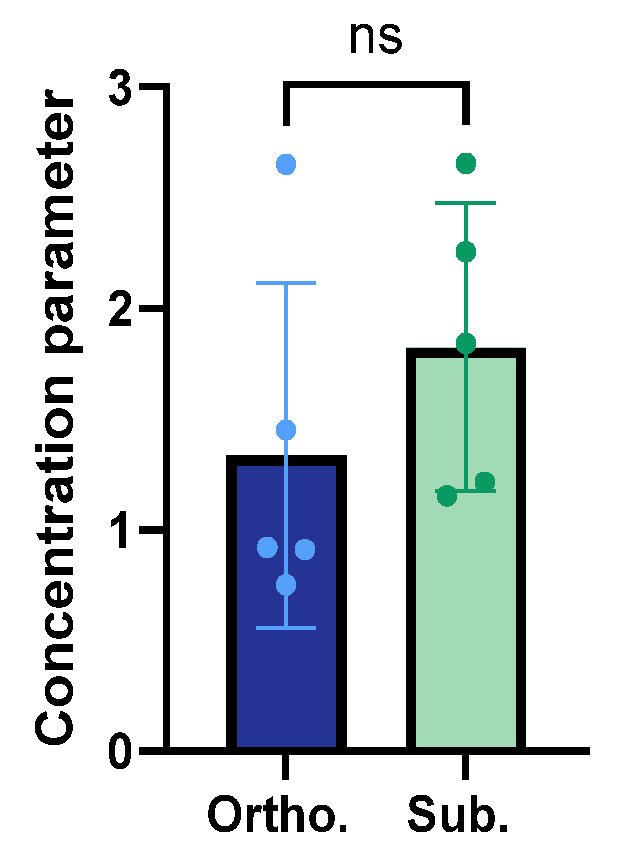
Distribution of the average concentration parameter $\bar{a}$ for the orthotropic and subcutaneous tumours. The height of the bar is the mean, and the error bar represents the SD. Each dot represents one tumour. The sample size was *n* = 5 for the orthotopic tumours and *n* = 5 for the subcutaneous tumours.

**Figure 5 cancers-15-05415-f005:**
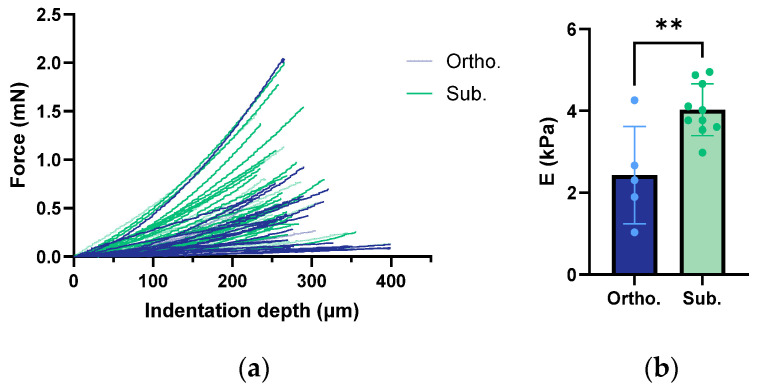
Indentation curves of the orthotopic and subcutaneous tumours. Each line represents one tumour (**a**). Young’s modulus (E) of the orthotopic and subcutaneous tumours was estimated (**b**). Each dot represents one tumour and an average of four to five measurements per tumour. The height of the bar is the mean, and the error bar represents the SD. The asterisks indicate statistically significant differences evaluated by a two-tailed *t*-test. (** *p* < 0.01). The sample size was *n* = 5 for the orthotopic tumours and *n* = 10 for the subcutaneous group.

**Figure 6 cancers-15-05415-f006:**
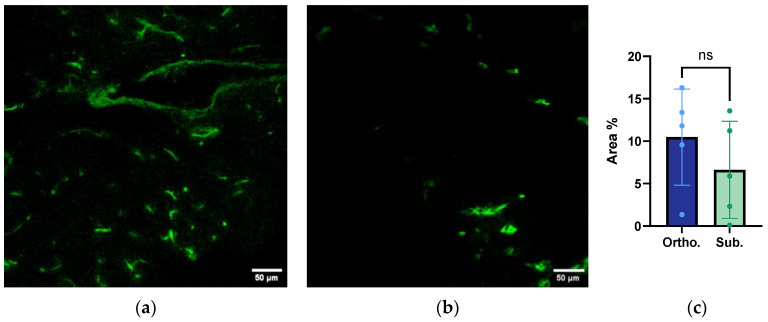
Confocal laser scanning microscopy (CLSM) of the lectin-FITC-labelled vessels of the orthotopic (**a**) and subcutaneous (**b**) KPC tumours. The scale bar represents 50 μm. (**c**) The quantitative analysis of the number of functional vessels imaged by CLSM of the orthotopic and subcutaneous KPC tumours. The area % represents the pixels corresponding to the lectin/FITC-labelled (functional) vessels per area. Each dot represents one tumour. The height of the bar is the mean, and the error bar represents the SD. The sample size was *n* = 5 for the orthotopic tumours and *n* = 10 for the subcutaneous group.

**Figure 7 cancers-15-05415-f007:**
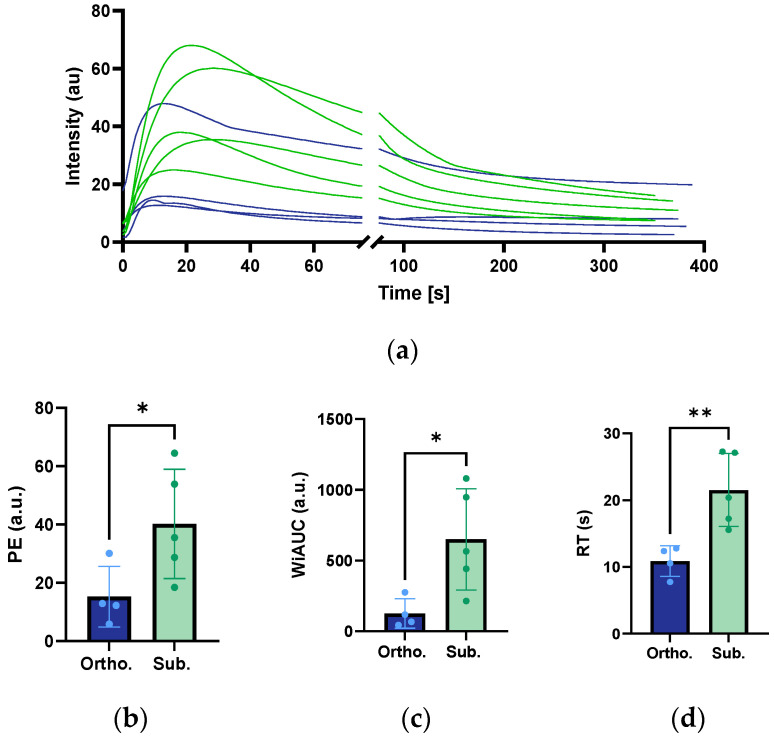
Time–intensity curves (**a**) of the orthotopic and subcutaneous KPC tumours. The orthotopic tumours are in blue, and the subcutaneous tumours are in green. Each line corresponds to one tumour. The sample size was *n* = 5 for the orthotopic tumours and *n* = 10 for the subcutaneous group. The peak enhancement (PE) of the orthotopic and subcutaneous KPC tumours is shown in (**b**) the wash-in area under the curve (WiAUC) in (**c**), and the rise time (RT) in (**d**). Each dot represents one tumour. The height of the bar is the mean, and the error bar represents the SD. The asterisks indicate statistically significant differences evaluated by a two-tailed *t*-test. (* *p* < 0.05, ** *p* < 0,01).

**Figure 8 cancers-15-05415-f008:**
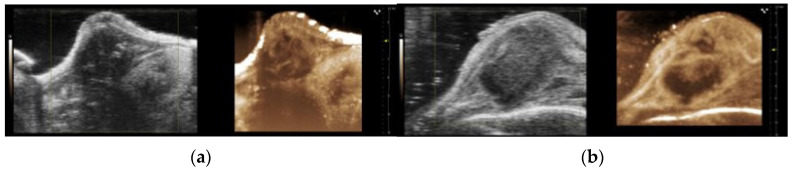
Representative B-mode and maximum intensity projections (MIP) based on the nonlinear contrast mode images for the orthotopic tumours (**a**) and the subcutaneous tumours (**b**).

**Figure 9 cancers-15-05415-f009:**
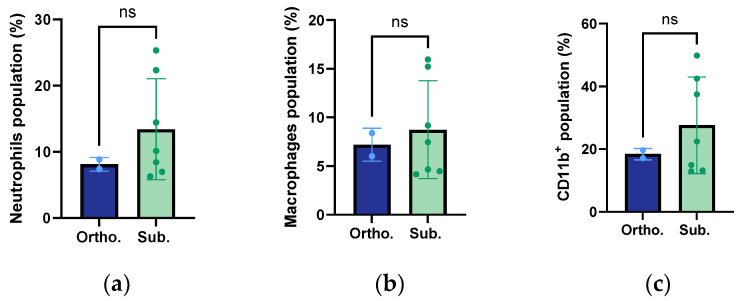
Mean percentage of neutrophils (**a**), macrophages (**b**) and innate immune cells (CD11b^+^) (**c**) in relation to the live cells of the orthotopic and subcutaneous KPC tumours. Each dot represents one tumour. The height of the bar is the mean, and the error bar represents the SD. The sample size was *n* = 2 for the orthotopic tumours and *n* = 7 for the subcutaneous group.

**Figure 10 cancers-15-05415-f010:**
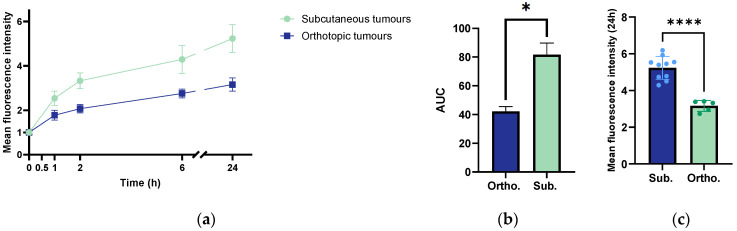
(**a**) Kinetics of the accumulation of 800CW as a function of time in the orthotopic and subcutaneous KPC tumours. The error bar represents the SD. (**b**) The area under the curve (AUC) corresponding to the curves in (**a**). The height of the bar is the mean, the error bar indicates a standard error, and the * indicates a statistical significance with a 95% confidence interval for the AUCs of 36 to 48 for the orthotopic model, and 66 to 98 for the subcutaneous model. (**c**) The accumulation of 800CW at the 24 h timepoint. The height of the bar is the mean, the error bar represents the SD, and the asterisks indicate statistically significant differences evaluated by a two-tailed *t*-test. (**** *p* < 0.0001). The sample size was *n* = 5 in the orthotopic group and *n* = 10 in the subcutaneous group.

**Figure 11 cancers-15-05415-f011:**
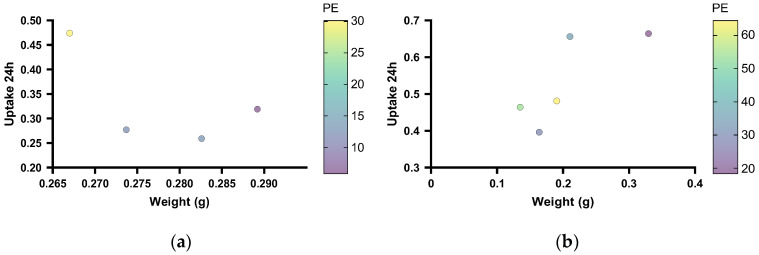
Multi-variable plots showing the correlation between the uptake of 800CW, the PE, and the weight of the orthotopic (**a**) and subcutaneous (**b**) KPC tumours.

**Table 1 cancers-15-05415-t001:** Metastases and infiltration.

Metastatic and Infiltration Events	Orthotopic Tumours	Subcutaneous Tumours
Infiltration	✓ (Adjacent tissue)	✓ (Skeletal muscle)
Metastasis in the spleen	✓	-
Metastasis in the intestines	✓	-
Metastasis in the kidneys	✓	-
Metastasis in the liver	✓	-

**Table 2 cancers-15-05415-t002:** CEUS parameters of the orthotopic and subcutaneous KPC tumours: the PE, WiAUC, RT, and area of the tumour cross-section (mm^2^). The parameters are presented as the mean value and SD.

CEUS Parameters	Orthotopic Tumours	Subcutaneous Tumours	*p*-Values
PE (a.u)	15.3 ±10.4	40.2 ± 18.7	0.050
WiAUC (a.u)	126.2 ± 103.8	650.0 ± 358.5	0.027
RT (s)	10.9 ± 2.3	21.5 ± 5.5	0.009
Tumour area (mm^2^)	74.7 ± 18.3	50.3 ± 7.9	0.029

**Table 3 cancers-15-05415-t003:** Summary of the results from the histopathological characterisation, tumour growth and metastases, collagen characterisation, Young’s modulus, functional vasculature, perfusion, immune cells, and accumulation of 800CW of the orthotopic and subcutaneous KPC tumours.

Characterisation	Orthotopic Tumours	Subcutaneous Tumours
Histopathological characterisation	Focal necrotic regions Glandular structures Infiltration of pancreatic tissue	Extended necrotic regions
Tumour growth and metastases	Larger volumes Higher growth rate Infiltration into healthy pancreatic tissue Metastasis to the spleen, intestines, kidneys, and liver.	Infiltration into skeletal muscle
Collagen characterisation	Lower amount of collagen Less aligned collagen fibres	Higher amount of collagen More aligned collagen fibres
Biomechanical characterisation	Less stiff tumours (Lower Young’s modulus)	Stiffer tumours (Higher Young’s modulus)
Functional vasculature	Trend—less functional microvasculature	Trend—more functional microvasculature
Perfusion Rise time	Smaller overall vascular volume Lower	Larger vascular volume Higher
Immune cells	Similar levels of neutrophils and macrophages	Similar levels of neutrophils and macrophages
Accumulation of 800CW	Lower and slower accumulation	Higher and faster accumulation

## Data Availability

The data presented in this study are available in article or Appendix A.

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
