# Peer review of "A Comparative Analysis of Orthotopic and Subcutaneous Pancreatic Tumour Models: Tumour Microenvironment and Drug Delivery"

_cancers, 2023, doi:10.3390/cancers15225415_

Round 1

Reviewer 1 Report

Comments and Suggestions for Authors

In this article, the authors investigated and compared the suitability of orthotopic and subcutaneous pancreatic ductal adenocarcinoma (PDAC) tumour models for drug delivery studies, evaluated various variables that characterise the tumour microenvironment, and conducted a comprehensive analysis of the structure and orientation of collagen in both tumour models. The results elucidate the impact of these factors on the uptake of a fluorescent model drug, 800CW, and suggest that while the orthotopic model offers a more clinically relevant microenvironment, the subcutaneous model demonstrates advantages in terms of drug uptake kinetics. The authors emphasized that this work is crucial for designing more clinically relevant experiments and improving our understanding of pancreatic cancer biology. However, there are several issues in this manuscript as outlined below.

1.       In Figure 1, all MT images of orthotopic and subcutaneous KPC tumors should be shown, with at least five per group. Similarly, in Figure 3 and 6, all SHIM or CLSM images of tumors should be shown, with at least five per group.

2.       In Figure 2, all B-mode images and ROIs of orthotopic and subcutaneous KPC tumors should be shown, with at least five per group. The corresponding results of all orthotopic and subcutaneous tumors, in terms of infiltration and metastatic sites, should be statistically listed in the table. Additionally, images depicting infiltration and metastatic sites should also be shown.

3.       The bar charts in Figure 4, 6 and 9 did not show statistical significance. If there is no statistical significance, nsshould be indicated..

4.       In Table 1, the P-values for each CEUS parameter of orthotopic and subcutaneous KPC tumors should be presented.

Comments on the Quality of English Language

Minor editing of English language required.

Reviewer 2 Report

Comments and Suggestions for Authors

This work investigated the differences between orthotopic and subcutaneous tumour models for the drug delivery. They include tumour growth, histopathology, functional vasculature, and structure and orientation of collagen. The result is OK. I recommend it for publication after carefully discussing the previously used models in this field. The abstract and introduction should be refined so that one can understand the novelty and importance of this work.

Reviewer 3 Report

Comments and Suggestions for Authors

The authors conducted a comparative analysis of biological properties of orthotopic and subcutaneous PDAC mouse models, which is novel and extends the knowledge of PDAC cancer biology. Although some details remain to be polished, there is no major flaw in this study. My comments will be as below.

Comments:

1.

A comprehensive table that summarizes and compares to show the similarity and difference of these two models should be added.

2.

A further discussion about how these results can help other researches design experiments based on these two models will add more significance of this study.

3.

A more detailed illustration (like arrows that point the specific cells or areas) and interpretation of the results and technical terms may help wide audiences to better understand the results.

4.

Regarding the drug uptake property of these two models, whether 800CW could represent the current used chemotherapeutic reagents?

5.

Since the topic is about tumor microenvironments (TME), it would be always intriguing to discuss about the infiltration of immune cells in the TME. An additional immunostaining against immune cells markers (such as CD8, CD4 and CD68) can be utilized to reveal whether difference exists between these two models.

6.

Some typesetting issues exist, such as a label over the image in Figure 10d.

Comments on the Quality of English Language

Minor editing of English language required

Round 2

Reviewer 3 Report

Comments and Suggestions for Authors

The manuscript has been substantially improved